# Insights into the Mechanism of Curaxin CBL0137 Epigenetic Activity: The Induction of DNA Demethylation and the Suppression of BET Family Proteins

**DOI:** 10.3390/ijms241612874

**Published:** 2023-08-17

**Authors:** Varvara Maksimova, Valeriia Popova, Anzhelika Prus, Evgeniya Lylova, Olga Usalka, Guzel Sagitova, Ekaterina Zhidkova, Julia Makus, Ekaterina Trapeznikova, Gennady Belitsky, Marianna Yakubovskaya, Kirill Kirsanov

**Affiliations:** 1Department of Chemical Carcinogenesis, Institute of Chemical Carcinogenesis, Blokhin National Medical Research Center of Oncology, 24 Kashirskoe Shosse, 115478 Moscow, Russia; lavarvar@gmail.com (V.M.); nuarrbio@gmail.com (V.P.); aikaprus2000@gmail.com (A.P.); e.s.lylova@gmail.com (E.L.); olgausalka@gmail.com (O.U.); zhidkova_em@mail.ru (E.Z.); ymakus@yandex.ru (J.M.); trapeznikova723@gmail.com (E.T.); belitsga@mail.ru (G.B.); kkirsanov85@yandex.ru (K.K.); 2Department of Biotechnology and Industrial Pharmacy, Lomonosov Institute of Fine Chemical Technologies, Russian Technological University (MIREA), 86 Vernadsky Avenue, 119571 Moscow, Russia; 3Institute of Clinical Medicine, Sechenov First Moscow State Medical University, 8-2 Trubetskaya Street, 119991 Moscow, Russia; guzelsag2015@gmail.com; 4Institute of Medicine, Peoples’ Friendship University of Russia, 6 Miklukho-Maklaya Street, 117198 Moscow, Russia

**Keywords:** CBL0137, epigenetic-targeted therapy, epigenetic modulators, DNA methylation, DNMT3A, BET family proteins, BRD4, BRD2, BRD3

## Abstract

The development of malignant tumors is caused by a complex combination of genetic mutations and epigenetic alterations, the latter of which are induced by either external environmental factors or signaling disruption following genetic mutations. Some types of cancer demonstrate a significant increase in epigenetic enzymes, and targeting these epigenetic alterations represents a compelling strategy to reverse cell transcriptome to the normal state, improving chemotherapy response. Curaxin CBL0137 is a new potent anticancer drug that has been shown to activate epigenetically silenced genes. However, its detailed effects on the enzymes of the epigenetic system of transcription regulation have not been studied. Here, we report that CBL0137 inhibits the expression of DNA methyltransferase DNMT3a in HeLa TI cells, both at the level of mRNA and protein, and it decreases the level of integral DNA methylation in Ca Ski cells. For the first time, it is shown that CBL0137 decreases the level of BET family proteins, BRD2, BRD3, and BRD4, the key participants in transcription elongation, followed by the corresponding gene expression enhancement. Furthermore, we demonstrate that CBL0137 does not affect the mechanisms of histone acetylation and methylation. The ability of CBL0137 to suppress DNMT3A and BET family proteins should be taken into consideration when combined chemotherapy is applied. Our data demonstrate the potential of CBL0137 to be used in the therapy of tumors with corresponding aberrant epigenetic profiles.

## 1. Introduction

Malignant neoplasms occur because of the accumulation of genetic abnormalities and epigenetic changes caused by either external environmental factors or internal signaling disruption [1]. The epigenetic regulation of transcription includes multiple mechanisms, generally categorized into three groups, namely DNA and RNA methylation, histone modifications, and noncoding RNA expression, all of which may be involved in carcinogenesis and tumor progression [1]. In particular, DNA methylation as well as histone methylation and acetylation are changed intensively in many types of tumors [2]. DNA methyltransferases (DNMTs), histone methyltransferases (HMTs), acetyltransferases (HATs) and deacetylases (HDACs), and the Bromodomain and Extra-Terminal Domain (BET) family proteins are the key enzymes of the abovementioned mechanisms [3]. The overexpression of particular enzymes leads to changes in epigenetic modification patterns and reprogramming of transcriptome, which increases oncogene expression [4]. Moreover, epigenetic changes were identified as one of the main factors of chemoresistance development [5]. The reversibility of chromatin modifications is the main feature of the epigenetic regulation of transcription, which forms a background to the development of epigenetic anticancer therapy [6]. Drug-induced epigenetic reprogramming represents a novel strategy to reverse or move the transcriptome of cancer cells to the normal state, preventing resistant clone formation and obtaining better clinical response [7]. Chromatin relaxation caused by epidrugs increases the efficacy of genotoxic cytostatics [8]. Thus, the properties of epigenetic agents can and should be used to improve combination anticancer therapy. Currently, the design and development of the epigenetic enzyme modulators, specifically, the inhibitors of DNMTs, HDACs), HMTs EZH2, G9a/GLP and DOT1L, and the inhibitors of BET family proteins , are of particular interest, as these enzymes are overexpressed in a wide number of cancers [9,10]. However, only eight epigenetic inhibitors have been approved for cancer treatment [11]. The development of new epigenetic drugs and further detailed analysis of their effects on epigenetic regulation is reasonable (or urgently expected).

CBL0137 [1,1′-(9-(2-(isopropylamino)ethyl)-9H-carbazole-3,6-diyl)bis(ethan-1-one)] is a low-molecular-weight nongenotoxic DNA-binding compound from the class of carbazoles referred to as the second generation of Curaxines [12]. CBL0137 possesses a broad spectrum of anticancer effects, which was confirmed in more than 50 preclinical studies [13,14,15,16,17]. Currently, CBL0137 is involved in Phase I-II clinical trials for the treatment of glioblastoma, melanoma, sarcoma, lymphoma, and some other types of cancer (NCT04870944, NCT03727789, and NCT05498792).

Kantidze et al. demonstrated that CBL0137 possesses chromatin remodeling activity [18]. This result is in agreement with the ability of CBL0137 to activate epigenetically silenced genes, which was shown in our previous studies on HeLa TI (HeLa Trichostatin-Induced) cells with the epigenetically silenced reporter gene *GFP* [19,20]. HeLa TI cell treatment with CBL0137 caused the significant activation of *GFP* expression similar to 5-azacytidine (5-Aza) or Trichostatin A (TSA). HeLa TI cell population was shown to possess a subset of more than 15 epigenetic factors participating in the reporter gene silencing [21,22]. These factors include enzymes from the following classes: DNMTs, HDACs, HMTs, and histone chaperones. The influence of CBL0137 on different components of epigenetic transcriptional regulation has never been studied; however, this aspect should be investigated to understand all the consequences of its use in clinical practice and, if specific epigenetic alterations occur in tumor cells, to target it properly.

Therefore, the main goal of our study was to analyze the effects of CBL0137 on the epigenetic regulation of gene expression via DNA methylation and histone modification mechanisms. Determining specific aims of our study we concentrated on the enzymes characterized as participating in epigenetic silencing in HeLa TI cells as epigenetic activity of CBL0137 was shown on that model system. In particular, to characterize the influence of CBL0137 on DNA methylation, we analyzed DNMT1 and DNMT3A enzyme levels and the expression levels of corresponding genes in HeLa TI cells treated with the drug. Since HeLa TI DNA is hypomethylated to demonstrate the consequences of CBL0137’s influence on global DNA methylation, we chose Ca Ski, a cell line of the same histology but with a higher level of DNA methylation. Additionally, we analyzed the changes in the level of DNA methylation after CBL0137 treatment in HT29 cells, in which DNA had a hypermethylated profile. Then, we analyzed CBL0137’s influence on histone modifications by evaluating the level of the integral acetylation of histone H3, as well as the expression and activity of HATs (HAT1, CBP, and P300) and HDACs (HDAC1 and HDAC3); the expression of the proteins from the BET family; the level of histone modifications H3K9me3, H3K27me3, and H4K20me3; and the expression of HMTs SUV39H1/H2, SUV420H1/H2, G9a, EZH2, and DOT1L.

## 2. Results

### 2.1. CBL0137 Inhibited DNMT1 and DNMT3A Expression

The integral epigenetic activity of CBL0137 was previously shown using the HeLa TI test system for the screening of epigenetically active compounds [20,23]. De novo DNA methyltransferase DNMT3a is one of the enzymes that implement the silencing of the *GFP* gene in HeLa TI cells. DNMT1 plays an important role in maintaining DNA methylation. Using Western blotting, we analyzed the effects of nontoxic concentrations of CBL0137 on the DNA methyltransferases DNMT1 and DNMT3A at the protein and mRNA levels in HeLa cells. We demonstrated that CBL0137 (1.2 μM and 0.6 μM) induced a 2.7- and 2-fold decrease in DNMT3A protein level after 8 h of incubation and a 4- and 1.9-fold decrease after 24 h of incubation, respectively (Figure 1A). RT-qPCR results demonstrated a 1.6- and 1.9-fold decrease in DNMT1 and DNMT3A mRNA levels after the treatment of the cells with CBL0137 in the concentration of 1.2 μM (Figure 1B).

### 2.2. CBL0137 Decreased the Level of DNA Methylation in Ca Ski Cells but Did Not Affect DNA Methylation in HT29 Cells

The level of DNA methylation in HeLa cells is rather low and thus does not allow for the analysis of the effect of CBL0137 on the integral level of DNA methylation [23]. Therefore, the cervical cancer cell line Ca Ski with a higher level of DNA methylation was selected for the model system for this study. Moreover, we used the colorectal cancer cell line HT29 with hypermethylated genomic DNA [24]. The amount of demethylated DNA was detected using methyl-sensitive restriction (MSRE) assay based on the intensity of the HpaII-digested genome DNA cleavage products, as described in Da Silva et al. [25]. We found a decrease in the intensity of HpaII-digested product by 33% after the treatment of Ca Ski cells with CBL0137 (0.5 μM), whereas there were no significant differences between experimental and control samples in HT29 cells after CBL0137 treatment (0.05 μM) (Figure 1C). Specific demethylation drug 5-Aza, used as a positive control, decreased the intensity of HpaII-digested products by 45% and 31% in Ca Ski and HT29 cells, respectively, thus highlighting the significant demethylating effect of CBL0137 in Ca Ski cells.

### 2.3. CBL0137 Did Not Affect the Level of Histone H3 Acetylation, as well as the Activity and Expression of Histone Acetyltransferases or HDAC1 and HDAC3 Expression

We studied the effects of CBL0137 in nontoxic concentrations of 0.6 μM and 1.2 μM on the integral H3 histone acetylation and the enzymatic regulators of this process in two steps. In the first step, we analyzed CBL0137’s effects on total H3 histone acetylation using antibodies to acetylated sites (K9 + K14 + K18 + K23 + K27). We observed no changes in the total H3 histone acetylation level after the CBL0137 exposure, while a two-fold increase was observed under the action of TSA (HDACs inhibitor) (Figure 2A).

The enzymatic activity of histone acetyltransferases was analyzed using Histone Acetyltransferase Activity Assay Kit. CBL0137 did not induce a significant increase in the activity of HAT enzymes (Figure 2B). Using RT-qPCR, we found no changes in *HAT1*, *CREBBP*, and *EP300* genes expression encoding transcription coactivators in HeLa TI cells after the treatment with CBL0137 in both concentrations (Figure 2C). The results of Western blotting of nuclear protein fraction did not demonstrate any effect on HDAC1 and HDAC3 expression (Figure 2D).

### 2.4. CBL0137 Inhibited the Expression of BET Family Proteins and Affected BRD2, BRD3 and BRD4 Gene Expression

We studied the effects of nontoxic concentrations of CBL0137 on mRNA and protein levels of BRD2, BRD3, and BRD4, ubiquitously expressed in all tissues and organs, and found a decrease in the level of all BET family proteins (Figure 3A).

CBL0137 (1.2 and 0.6 μM) decreased BRD2 expression in HeLa TI cells after 4 h incubation by 2.1- and 4-fold, respectively. After 8 h of incubation, the BRD2 protein expression level decreased by 2.1- and 2.3-fold, and after 24 h of incubation, BRD2 expression was restored to the control level. BRD3 protein expression was decreased after the treatment with CBL0137 at both concentrations at each time point: a 2- and 1.4-fold decrease at 4 h, a 1.8- and 1.7-fold decrease at 8 h, and a 2.2- and 2.1-fold decrease at 24 h. We also observed a decrease in BRD4 protein expression after the treatment of HeLa TI cells with 1.2 and 0.6 μM CBL0137 at each time point: a 2-fold decrease after 4 h of incubation for both concentrations, a 1.6- and 2.5-fold decrease after the 8 h, and a 1.6- and 1.7-fold decrease after the 24 h, respectively.

Using RT-qPCR, we found an increase in *BRD2* gene expression by 3.4- and 1.6-fold after 24 h of treatment with 1.2 μM and 0.6 μM of CBL0137, respectively (Figure 3B). *BRD3* gene expression was increased by 2.2-fold after the 8 h of incubation with CBL0137 (1.2 μM) and by 2.9-fold after 24 h. Notably, 4 h incubation of the cells with CBL0137 in the concentrations of 1.2 μM and 0.6 μM led to a weak but reproducible decrease in *BRD4* expression (by 1.4- and 1.3-fold, respectively). However, after the 8 h of treatment, *BRD4* expression was increased by 1.3- and 1.7-fold, correspondingly. In 24 h of incubation, the increase in *BRD4* expression was 1.4-fold in the case of both CBL0137 concentrations.

### 2.5. CBL0137 Did Not Affect the Integral Level of H3K9me3, H3K27me3, and H4K20me3 Histone Modifications in HeLa TI Cells

Furthermore, we analyzed the effects of CBL0137 on the level of key histone H3 and H4 modifications associated with the repression of transcription, specifically, H3K9me3, H3K27me3, and H4K20me3, using Western blotting, and observed no significant changes (Figure 4A).

### 2.6. CBL0137 Did Not Affect SUV39H1/H2, SUV420H1/H2, G9a, EZH2, and DOT1L Expression Levels

We evaluated CBL0137’s effects on the enzymes responsible for H3 and H4 histone methylation. We selected the following enzymes as putative targets: SUV39H1 and SUV39H1 (di- and trimethylation of H3K9), G9a (mono- and dimethylation of H3K9), EZH2 (methylation of H3K27), SUV420H1 and SUV420H2 (di- and trimethylation of H4K20), and DOT1L (methylation of H3K79). The protein expression level was detected via Western blotting. We observed no significant changes in SUV39H1, SUV39H2, and G9a protein expression levels after 24 h of the treatment of the cells with 1.2 μM and 0.6 μM of CBL0137 (Figure 4B). We also did not detect any changes in EZH2, SUV420H1, SUV420H2, and DOT1L expression.

## 3. Discussion

Curaxin CBL0137 is considered to be a multitarget anticancer drug affecting spatial genome organization. The main mechanism of CBL0137 action is considered to be DNA binding, followed by nucleosome destabilization and the inhibition of protein complex FACT (FAcilitates Chromatin Transcription). CBL0137 also causes the dissociation of CTCF from its binding sites, the disruption of chromatin loops, and the destruction of 3D genome organization [12,26,27,28]. CBL0137 represses the enhancer- and super-enhancer-activated transcription, which are critical for oncogene expression, in particular, the expression of c-MYC family oncogenes [26]. FACT inhibition negatively regulates a number of metabolic signaling pathways involved in cancer development [27]. Our study contributes to the knowledge of the influence of CBL0137 on the mechanisms of the epigenetic regulation of gene expression, in particular, DNA methylation, the methylation and acetylation of H3 and H4 histones, and BET family enzyme functioning.

Previously, it was demonstrated that CBL0137 induced significant reactivation of the expression of epigenetically repressed *GFP* gene in HeLa TI cells [19]. DNMT3A methyltransferase is 1 out of the 15 factors carrying out the silencing in HeLa TI cells. This result is in agreement with data of Poleshko et al., concerning the involvement of DNA methylation in *GFP* repression [21]. Besides that, in a previous study using HeLa TI cells, we demonstrated the overlapping effects of CBL0137 and demethylating agent 5-Aza in transcription reactivation, which is possibly due to the impact of some similar mechanisms in the action of both agents [20]. The chemical structure of CBL0137 mediates its ability to bind DNA. The carbazole core intercalates between DNA bases with symmetrical side chains protruding into major grooves, while the N-side carbazole chain lies into minor grooves [29]. In the present study, we proved the hypothesis of the effects of CBL0137 on DNA methylation. For the first time, we showed that CBL0137 significantly inhibited DNMT3A enzyme expression. CBL0137 also decreased the level of methylated DNA in Ca Ski cells but not in HT29 cells. This result could be explained by the different levels of genomic DNA methylation in cell lines. Previously Sergeev et al. observed a similar CBL0137 effect in the cell-free system in vitro. Based on the ability of CBL0137 to intercalate into DNA, the authors highlighted the potential of CBL0137 to disrupt the DNMT3a-induced DNA methylation due to the changes in the affinity between DNA and methyltransferase. CBL0137 in the range of concentration of 5–30 μM decreases DNMT3a binding with DNA, and in the range of concentration of 10–60 μM, it decreases the efficacy of DNA methylation [30]. CBL0137 revealed remarkable anticancer activity in vitro and in vivo in the model of acute myeloid leukemia (AML) [13]. This malignancy is characterized by DNMT3A mutations, leading to changes in the DNA methylation profile [31]. Notably, DNMTs inhibitor 5-Aza is effective in AML therapy [32]. Thus, the anticancer efficacy of CBL0137 in hematological malignancies may be partially mediated by the inhibition of DNMT3A, but this mechanism needs further investigation.

Various functional states of the enhancer landscape can be characterized by specific combinatorial patterns of histone modifications. Histone acetylation is the mechanism involved in the activation of super-enhancers [33]. In our study, we analyzed the effect of CBL0137 on the integral histone acetylation and evaluated its effect on the components of acetylation signaling, in particular, histone acetyltransferases and histone deacetylases. For the first time, we evaluated the effect of CBL0137 on HAT enzymes. The obtained results confirmed the absence of any effects on the enzymatic activity of HAT and the expression of *HAT1*,* CREBBP*, and *EP300* genes. CBL0137’s effect on the H3 histone integral acetylation was also not observed. The obtained data correlate with the results of Leonova et al., using the same cells but with lower CBL0137 concentration. It should be noted that we analyzed five sites of H3 histone acetylation (K9, K14, K18, K23, and K27). Zhou et al. demonstrated an increase in H3K9ac modification level in normal HEK293T cells after incubation with CBL0137 at concentrations of 0.1, 0.2, and 0.5 μM. In addition, a decrease in the phosphorylated form of HDAC1 with no effect of CBL0137 on HDAC1 expression was shown [34]. In this study, we also found no effect of CBL0137 on HDAC1 and HDAC3 protein levels. We also did not detect any CBL0137 effect on the level of HDAC1 and HDAC3 protein levels. BET family proteins are another important component of transcription epigenetic regulation. The main members of this family are four bromodomain-containing proteins: BRD2, BRD3, and BRD4, expressed in all mammalian cells and tissues; and BRDT, specifically expressed in testicles [35]. BET family proteins bind to hyperacetylated sites of the histones, and they are accumulated in transcriptionally active chromatin [36]. However, BET family proteins can also interact with nonhistone proteins, and their role goes beyond histone acetylation recognition [37]. For the first time, we revealed that CBL0137 inhibits the expression of BRD2, BRD3, and BRD4 proteins. Notably, the inhibition of protein expression was not associated with a decrease in mRNA level, pointing to the effect of CBL0137 on the protein degradation system. The fact that the decrease in the levels of BET family proteins was observed earlier than the increase in mRNA, which was also time- and dose-dependent, may be interpreted as the observed downregulation. BRD4 plays a significant role at various stages of the transcription process [38]. The main function of BRD4 is to recruit the transcription elongation factor P-TEFb into the promoter–proximal region of genes [35]. The role of BRD2 and BRD3 proteins is less studied, but it is known that they participate in Pol II initiation in combination or individually [35]. The BRD2-enriched chromatin regions also include the histone variant H2A.Z, regulating both the initiation and elongation of Pol II and the deployment of the nucleosome [39].

The decrease in the expression of BET family proteins is likely related to the effect of CBL0137 on the FACT protein complex. FACT consists of SSRP1 and SUPT16H subunits. Zhou et al. demonstrated that BRD4 plays a key role in the reading of SUPT16H acetylation. Besides that, the chemical inhibition of BRD4 by JQ-1 and UMB-136 compounds was shown to cause a significant decrease in SUPT16H expression, which is in agreement with the effect of CBL0137 [26,34]. BRD4 is a key mediator of YAP/TAZ transcriptional coactivators regulating the activity of a large set of enhancers with super-enhancer functional properties [40]. BRD4 is involved in the regulation of various oncogenes responsible for the control of cell proliferation, resistance to apoptosis, and aggressiveness [38]. The main targets of BRD4-regulated enhancers are *MYC*, *FOS*, and *AURKB* genes [41]. Thus, we found the effects of CBL0137 to be similar to the BET inhibitors competing with BRD proteins in promoter and enhancer regions and disrupting BRD-mediated transcription regulation. Despite the fact that the mechanism of the direct interaction of CBL0137 with BRD proteins remains unknown and requires further investigation, CBL0137 can be considered a potential BET inhibitor.

Our results concerning CBL0137 effects on DNMT3A and BET family proteins are supported by recent data from Xiao et al., who showed a synergistic effect of CBL0137 and Panobinostat, an FDA-approved HDAC inhibitor, in a *MYCN*-amplified neuroblastoma model [42]. The results of this study are consistent with those of Shahbazi et al., who demonstrated synergism between the BRD4 inhibitor JQ-1 and Panobinostat in the same model [43]. The use of a combination of CBL0137 with HDAC inhibitors can be considered a reasonable approach for combined chemotherapy with epigenetic drugs.

A number of methylated histone modifications are necessary for the maintenance of the chromatin structure [44]. Consequently, the destabilization of nucleosomes caused by CBL0137 may lead to global changes in histone methylation. However, we demonstrated the absence of the CBL0137 effect on H3K9me3, H3K27me3, and H4K20me3, which are critical for heterochromatin formation. However, Zhou et al. observed a dose-dependent decrease in H3K9me3 and H3K27me3 modification levels after the treatment of HEK293T cells with CBL0137 [34]. HEK293T cells are likely more flexible in epigenetic changes compared with HeLa TI cells. The level of H4K20me3 modification after the treatment of the cells with CBL0137 was also analyzed for the first time. The HMTs expression data are in agreement with the results of the integral analysis of histone methylation. We did not observe any changes in the level of SUV39H1, SUV39H2, and G9a proteins in HeLa TI cells after the treatment with CBL0137. Similar results were obtained for the EZH2 protein, associated with the modification of H3K27me3, as well as SUV420H1 and SUV420H2 proteins, responsible for the trimethylation of H4K20. Also, no effects were observed on the DOT1L enzyme. To date, the effect of CBL0137 on these methyltransferases has not been studied.

## 4. Materials and Methods

### 4.1. Cell Cultivating

HeLa TI is a polyclonal population of the HeLa cell line (the epidermoid carcinoma of the cervix) carrying a genome-generated vector based on the avian sarcoma virus and driven by an epigenetically repressed *GFP* gene; Ca Ski is a cell culture of the epidermoid carcinoma of the cervix with a higher level of DNA methylation than HeLa cells [23]. The HT29 cell line comprises colorectal adenocarcinoma cells. HeLa TI, Ca Ski, and HT29 cells were maintained in Dulbecco’s Modified Eagle’s Medium (DMEM) supplemented with 4.5 g/liter glucose (PanEco, Moscow, Russia), 10% (*v*/*v*) heat-inactivated fetal bovine serum (Biosera, Cholet, France), antibiotic cocktail containing penicillin (50 units/mL) and streptomycin (50 µg/mL) (PanEco, Moscow, Russia), and 2 mM L-glutamine (PanEco, Moscow, Russia). The cells were cultured under standard conditions (37 °C, 5% CO_2_). The cells were obtained from the nitrogen bank of Blokhin National Medical Research Center of Oncology.

### 4.2. MTT Test

Ca Ski cells were seeded in 96-well plates (SPL Lifescience, Pochon, Republic of Korea) at 5 × 10^3^ cells per well. The cells were treated with 16 serial dilutions of CBL0137 (0.02 μM—0.8 mM). All treatments were performed in triplicate. The cells were incubated with CBL0137 for 72 h (37 °C, 5% CO_2_). After incubation, the MTT reagent (0.25 mg/mL) (Alfa Aesar, Karlsruhe, Germany) was added to the plate and incubated for 4 h (37 °C, 5% CO_2_). After incubation, the medium was removed and DMSO (PanEco, Moscow, Russia) was added to dissolve the formazan crystals. Changes in optical density were detected at a wavelength of 540 nm using a spectrophotometer Multiskan SkyHigh (Thermo Fisher Scientific Inc., Waltham, MA, USA). Using the method of nonlinear regression, nontoxic concentrations of CBL0137 were determined. The experiment was carried out at least five times. For HT29 and HeLa TI cells, we used the nontoxic concentrations determined in previous studies [13,19].

### 4.3. DNA Extraction

Ca Ski and HT29 cells were seeded in 6-well plates (SPL Lifescience, Pochon, Republic of Korea) at 3 × 10^5^ cells per well and incubated overnight under standard conditions. Next, the cells were treated with CBL0137 (0.5 µM and 0.05 µM) and 5-Aza (1 µM, positive control) or DMSO (0.1%) as vehicle control. Every 24 h, half of the medium was replaced, and CBL0137 was added to the initial concentration. After 72 h of incubation, genomic DNA was extracted from the cells using the GeneJET Genomic DNA Purification Kit (K0721, Thermo Fisher Scientific Inc., Waltham, MA, USA). The concentration of extracted DNA was determined with a NanoDrop Lite spectrophotometer (Thermo Fisher Scientific Inc., Waltham, MA, USA).

### 4.4. MSRE Assay

MSRE assay was used for the detection of demethylated C^CGG sites after CBL0137 treatment. The enzymatic reaction with HpaII and MspI restriction enzymes was performed using EpiJET DNA Methylation Analysis Kit (K1441, Thermo Fisher Scientific Inc., Waltham, MA, USA) according to the manufacturer’s protocol. Briefly, 0.5 µg of genomic DNA was digested with Epi HpaII or Epi MspI, or not digested. HpaII partially digested genomic DNA (it is sensitive only to the hemimethylated C^CGG sequence), while MspI is methylation-insensitive and is cleaved both fully methylated, hemimethylated, and unmethylated sequences (this enzyme acts as a control of the reaction inhibition). Restriction products were analyzed using 1% agarose gel electrophoresis with TAE buffer and 1 kb Plus DNA Ladder (Evrogen, Moscow, Russia) and were detected on a Typhoon 9400 scanner (GE Healthcare, Chicago, IL, USA). The densitometric analysis of HpaII-digested DNA intensity was performed using ImageJ version 1.53 and Microsoft Excel version 2208. All experiments were performed at least three times.

### 4.5. Protein Extraction

#### 4.5.1. Histones

HeLa TI cells were seeded in 60 mm Petri dishes (SPL Lifescience, Pochon, Republic of Korea) at 3 × 10^5^ cells per well and incubated overnight under standard conditions. The cells were incubated with CBL0137 (0.6 µM and 1.2 µM), HDACs inhibitor TSA (0.25 μM, positive control), or DMSO (0.1%) as vehicle control for 24 h; then, they were washed with sodium phosphate buffer (PBS) (Biomed, Saint-Petersburg, Russia). Histones were extracted using the “Histone Extraction Protocol” (Abcam, Boston, MA, USA). Briefly, the cells were lysed with Triton Extraction Buffer buffer (PBS containing 0.5% (*v*/*v*) Triton X-100 (Ferak, Berlin, Germany), 2 mM PMSF (Sigma-Aldrich, Darmstadt, Germany), and 0.02% (*v*/*v*) NaN3 (Sigma-Aldrich, Darmstadt, Germany)) supplemented with Complete Protease Inhibitor Cocktail (Roche, Basel, Switzerland). Then, histones were extracted with 0.2 N HCl and reprecipitated with TCA (instead of neutralization with NaOH) as per the published protocol [45].

#### 4.5.2. Total Protein Fraction

HeLa TI cells were seeded in 6-well plates (SPL Lifescience, Pochon, Republic of Korea) at 2 × 10^5^ cells per well and incubated overnight under standard conditions. The cells were incubated with CBL0137 (0.6 µM and 1.2 µM) or DMSO (0.1%) as vehicle control for 4, 8, and 24 h. Then, they were washed with PBS (Biomed, Saint-Petersburg, Russia) and lysed with RIPA buffer (150 mM NaCl (PanEco, Moscow, Russia), 1% (*v*/*v*) Triton X-100 (Ferak, Berlin, Germany), 0.5% (*w*/*v*) sodium deoxycholate (AppliChem, Darmstadt, Germany), 0.1% (*w*/*v*) Sodium dodecyl sulfate (SDS, SERVA, Heidelberg, Germany), and 50 mM Tris-HCl pH 8.0 (PanEco, Moscow, Russia)). The cells were lysed with constant stirring at 4 °C for 1 h. Next, the samples were centrifuged (1790× *g*, 5 min, 4 °C), and the supernatant containing the total protein fraction was collected. Protein concentration was measured using the Bradford method [46].

#### 4.5.3. Nuclear Protein Fraction

Cells were seeded in 60 mm Petri dishes (SPL Lifescience, Pochon, Republic of Korea) at 3 × 10^5^ per dish and incubated overnight under standard conditions. Afterward, the cells were treated with CBL0137 at concentrations of 0.6 μM and 1.2 μM or DMSO (0.1%) as vehicle control. After 24 h, the cells were washed with PBS and lysed in buffer for nuclear extraction and fractionation (Abcam protocol: 20 mM HEPES pH 7.4 (PanEco, Moscow, Russia), 10 mM KCl (CHIMMED, Moscow, Russia), 2 mM MgCl_2_ (CHIMMED, Moscow, Russia), 1 mM EDTA (PanEco, Moscow, Russia), 1 mM EGTA (AppliChem, Darmstadt, Germany), 20 mM dithiothreitol (SERVA, Heidelberg, Germany), and Complete Protease Inhibitor Cocktail (Roche, Basel, Switzerland)). Briefly, the cells were lysed for 15 min on ice, and then cell suspension was performed through a 26- and 23-gauge needle with the removal of supernatant (for the mechanical destruction of the cytoplasmic and nuclear membrane, respectively). The precipitate was resuspended in deionized water with Complete Protease Inhibitor Cocktail (Roche, Basel, Switzerland) and then homogenized with Soniprep 150 Plus ultrasonic homogenizer (MSE, London, UK) for 3 s at a power of 2A on ice. The nuclear fraction was used to analyze the enzymatic activity of HATs as well as the level of HDAC1 and HDAC3 protein expression.

### 4.6. Western Blotting

#### 4.6.1. Histone Modifications

Proteins were separated using 15% SDS-PAGE with Tris-Gly buffer (25 mM Tris, 189 mM Glycine (both PanEco, Moscow, Russia)) and transferred to 0.22 μm nitrocellulose membrane (Bio-Rad, Hercules, CA, USA) for 40 min at 100 mA on ice. Membrane blocking was performed using 2.5% nonfat milk in TBST (45 min, 4 °C). To assess the quality of protein transfer, the membrane was stained in Ponceau S solution (0.2% (*w*/*v*) Ponceau S in 3% acetic acid). For the primary hybridization, rabbit antibodies (Abcam, Cambridge, UK) to panH3ac (ab47915, 1:5000), H3K9me3 (ab8898, 1:3000), and H4K20me3 (ab9053, 1:5000) were used. Goat antibodies (Abcam, Cambridge, UK, ab97051, 1:10,000) were used as secondary antibodies. To control protein loading, the membranes were stripped with buffer (0.06M Tris-HCl pH 6.8 (PanEco, Moscow, Russia), 0.8% (*v*/*v*) 2-mercaptethanol (Sigma-Aldrich, Darmstadt, Germany), and 0.01% (*v*/*v*) SDS (SERVA, Heidelberg, Germany)) and then hybridized with rabbit antibodies to H4 (Abcam, Cambridge, UK, ab10158, 1:3000). Proteins were detected using Clarity™ Western ECL Substrate (Bio-Rad, Hercules, CA, USA) and ImageQuant LAS 4000 Digital Imaging System (GE Healthcare, Chicago, IL, USA). The densitometric analysis of the blots was performed using ImageJ version 1.53 and Microsoft Excel version 2208 according to the protocol of Li et al. [47]. All experiments were performed at least three times.

#### 4.6.2. DNMTs, HMTs, and BET Family Proteins

Total fraction proteins were separated using 10% SDS-PAGE with Tris-Gly buffer and transferred to 0.45 μm nitrocellulose membrane (Bio-Rad, Hercules, CA, USA) for 1 h at 250 mA on ice. The assessment of the quality of protein transfer, hybridization with primary and secondary antibodies, and protein development were performed as described previously. For primary hybridization, rabbit antibodies (Abcam, Cambridge, UK) to SUV39H1 (ab245380, 1:3000), SUV39H2 (ab229493, 1:3000), G9a (ab183889, 1:3000), DOT1L (ab64077, 1:500), EZH2 (ab228697, 1:7000), BRD2 (ab139690, 1:10,000), BRD3 (ab264294, 1:5000), BRD4 (ab128874, 1:1000), and DNMT3a (ab2850, 1:500); rabbit antibodies (Invitrogen, Carlsbad, CA, USA) SUV420H1 (PA5-40926, 1:3000) and SUV420H2 (PA5-109891, 1:3000); and rabbit antibodies DNMT1 (ABclonal, San Diego, CA, USA, A5495, 1:3000) were used. Goat antibodies (Abcam, ab97051, 1:10,000) were used for secondary hybridization. To control protein loading, rabbit antibodies to β-actin (Abcam, Cambridge, UK, ab227387, 1:5000) were used. All experiments were performed at least three times.

#### 4.6.3. HDACs

Nuclear fraction proteins were separated using 10% SDS-PAGE with Tris-Gly buffer. Next, proteins ranging in size from 25 to 75 kDa were transferred to a 0.45 μm nitrocellulose membrane (Bio-Rad, Hercules, CA, USA) for 1 h at 200 mA. Proteins less than 25 kDa were transferred to a 0.22 μm nitrocellulose membrane (Bio-Rad, Hercules, CA, USA) in a separate sandwich. The 0.22 µm membranes were removed 30 min after the start of the transfer. The quality of protein transfer, hybridization with primary and secondary antibodies, and protein development were assessed as described previously. For primary hybridization, rabbit antibodies (Abcam, Cambridge, UK) HDAC1 (ab7028, 1:3000) and HDAC3 (ab32369, 1:5000) were used. Goat antibodies (Abcam, Cambridge, UK, ab97051, 1:10,000) were used for secondary hybridization. To control protein loading, rabbit antibodies to histone H3 (Abcam, Cambridge, UK, ab18521, 1:3000) were used. All experiments were performed at least three times.

### 4.7. HAT Activity Assay

Nuclear HAT activity was quantified using Histone Acetyltransferase Activity Assay Kit (ab65352, Abcam, Boston, MA, USA) according to the manufacturer’s instructions. Briefly, 50 μg of nuclear extract was incubated with HAT I/II and NADH-generating enzyme in HAT assay buffer for 4 h at 37 °C. The optical density was measured at 450 nm in a Multiskan Sky microplate spectrophotometer (Thermo Fisher Scientific Inc., Waltham, MA, USA). Active nuclear extract was used as a positive control. HAT activity was expressed as the relative O.D. value per μg, after which the activity was calculated in % relative to the vehicle control. All experiments were performed in technical triplicate at least three times.

### 4.8. RNA Extraction and cDNA Synthesis

HeLa TI cells were seeded in 6-well plates (SPL Lifescience, Pochon, Republic of Korea) at 2 × 10^5^ cells per well. Cells were incubated with CBL0137 (0.6 µM and 1.2 µM) for 4, 8 and 24 h. Then, the total RNA was extracted using ExtractRNA reagent (Evrogen, Moscow, Russia) according to the manufacturer’s protocol. After total RNA extraction, the samples were treated with DNase I (Syntol, Moscow, Russia), and then RNA reprecipitation with ExtractRNA was performed. The concentration of extracted RNA was determined with a NanoDrop Lite spectrophotometer (Thermo Fisher Scientific Inc., Waltham, MA, USA). For cDNA synthesis, 2 μg of total RNA was reverse-transcribed using MMLV-RT (SK022L, Evrogen, Moscow, Russia) and random (dN)10 primers in a 25 μL reaction volume following the manufacturer’s protocol.

### 4.9. Reverse-Transcription Quantitative PCR

RT-qPCR was carried out using the 5X qPCRmix-HS SYBR Kit (PK147L, Evrogen, Moscow, Russia) according to the manufacturer’s protocol in 96-well plates (Kirgen Bioscience, Shanghai, China). Forward and reverse primers were used at a concentration of 0.2 μM. The sequences used for amplification are presented in Table 1. RT-qPCR was performed using a CFX96 Connect cycler (Bio-Rad, Hercules, CA, USA) with the following thermal cycles: initial denaturation step by heating at 95 °C for 5 min, followed by 40 cycles of 20 s at 95 °C (denaturation), 20 s at 58 °C (annealing), and 25 s at 72 °C (elongation). The expression of the gene of interest was normalized to that of the housekeeping gene *RPLP0*. The changes in the levels of mRNA expressions of the studied genes were calculated using the 2^−^^ΔΔCt^ method [48]. All experiments were performed in technical triplicate at least three times.

### 4.10. Statistical Data Analysis

Statistical data analysis was performed with the GraphPad Prism 8.3.0 software. The normality of data distribution was assessed with the Kolmogorov–Smirnov test. The significance of the difference in the effects of CBL0137 relative to vehicle control in MSRE assay and Western blot was analyzed using one-way ANOVA with the Fisher LSD test, and in RT-qPCR, it was assessed using two-way ANOVA with the Fisher LSD test. In all cases, the differences were considered significant at *p* < 0.05.

## 5. Conclusions

In the present study, we performed the screening of CBL0137’s effect on DNA methylation as well as on the acetylation and methylation of H3 and H4 histones. For the first time, we demonstrated CBL-induced inhibition of DNMT3A expression at the protein and mRNA levels, as well as the inhibition of the expression of bromodomain-containing proteins of the BET family, BRD2, BRD3, and BRD4. Notably, CBL0137 does not influence the mechanisms of histone acetylation and methylation.

These findings highlight the potential of CBL0137 as a therapeutic option for tumors characterized by DNMT and BET overexpression. The utilization of CBL0137 could provide innovative approaches to overcome therapeutic resistance and improve the efficacy of anticancer treatments. A complete understanding of the mechanisms and therapeutic implications of CBL0137-mediated epigenetic regulation is essential in order to build the most effective strategy for the treatment of tumors with epigenetic disorders. CBL0137 can become an important component of combined anticancer therapy.

## Figures and Tables

**Figure 1 ijms-24-12874-f001:**
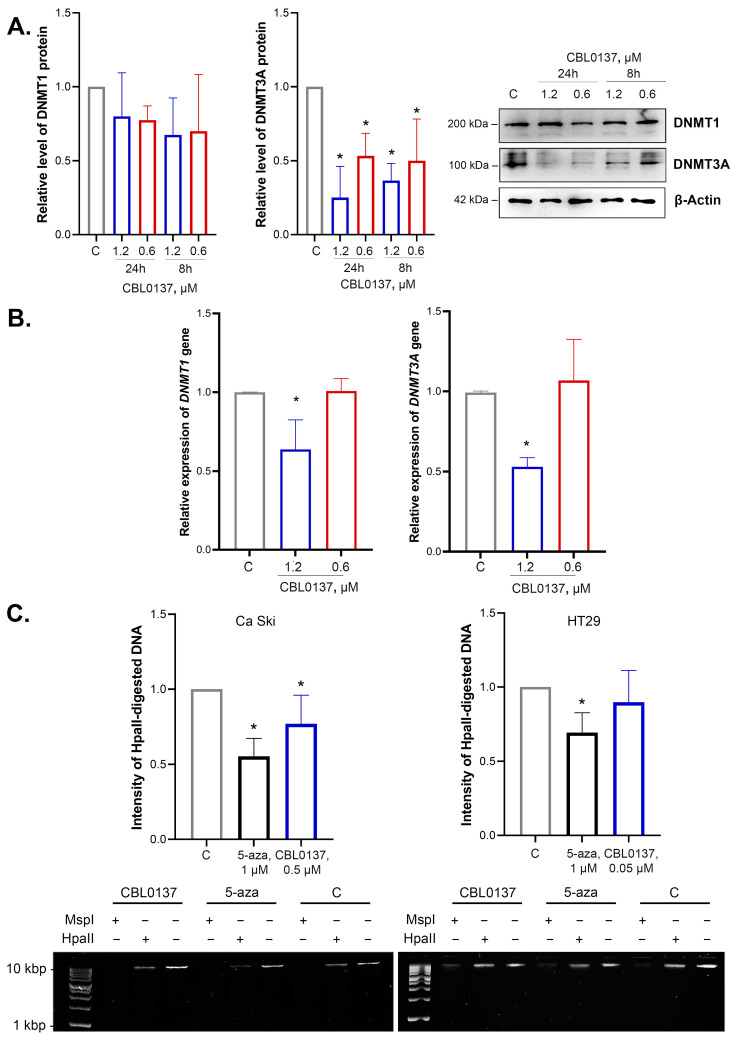
CBL0137 reduced the expression of DNMT3A at the protein and mRNA levels and inhibited the DNMT1 gene expression and induced DNA demethylation: (**A**) DNMT1 and DNMT3A protein expressions were assessed via Western blotting after treatment of HeLa TI cells with CBL0137 (1.2 and 0.6 μM) for 24 h and 8 h. The results were normalized to β-actin and are shown as mean ± SD of the densitometric analysis (n ≥ 3); * *p* < 0.05 compared to vehicle control (C). (**B**) DNMT1 and DNMT3A gene expressions were assessed with RT-qPCR after treatment of HeLa TI cells with CBL0137 (1.2 and 0.6 μM) for 24 h. The results were normalized to the housekeeping gene RPLP0 and are shown as mean ± SD (n ≥ 3); * *p* < 0.05 compared to vehicle control (C). The 2^−ΔΔCt^ method was used for data calculation. (**C**) The degree of DNA demethylation was evaluated with MSRE after treatment Ca Ski and HT29 cells with DMSO (0.1%, vehicle control), CBL0137 (0.5 μM and 0.05 μM, respectively), or 5-Aza (1 μM, used as positive control) for 72 h. * *p* < 0.05 compared to vehicle control (C).

**Figure 2 ijms-24-12874-f002:**
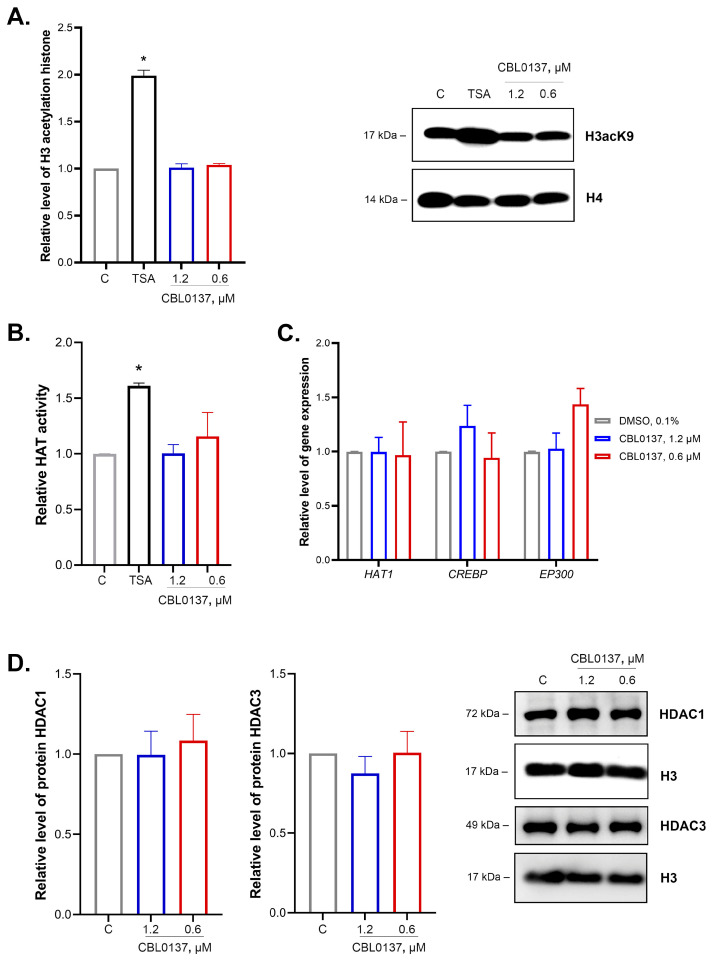
CBL0137 did not affect the acetylation and expression of H3 histone, the activity of histone acetyltransferases, or HDAC1 and HDAC3 expression: (**A**) Histone H3 acetylation was assessed using Western blotting after treatment of HeLa TI cells with CBL0137 (1.2 and 0.6 μM) or TSA (0.25 μM, used as positive control) for 24 h. Western blotting results were normalized to histone H4 and are shown as mean ± SD of the densitometric analysis (n ≥ 3); * *p* < 0.05 compared to vehicle control (C). (**B**) Enzymatic activity of the HAT family proteins was assessed using Histone Acetyltransferase Activity Assay Kit after treatment of HeLa TI cells with CBL0137 (1.2 and 0.6 μM) for 24 h. The results are shown as mean ± SD (n ≥ 3); * *p* < 0.05 compared to vehicle control. (**C**) *HAT1*, *CREBBP*, and *EP300* gene expressions were assessed with RT-qPCR after treatment of HeLa TI cells with CBL0137 (1.2 and 0.6 μM) for 24 h. The results were normalized to the housekeeping gene *RPLP0* expression and are shown as mean ± SD (n ≥ 3). The 2^−ΔΔCt^ method was used for data calculation. (**D**) Expressions of HDAC1 and HDAC3 were assessed using Western blotting after treatment of HeLa TI cells with CBL0137 (1.2 and 0.6 μM) for 24 h. The results were normalized to β-actin and are shown as mean ± SD of the densitometric analysis (n ≥ 3).

**Figure 3 ijms-24-12874-f003:**
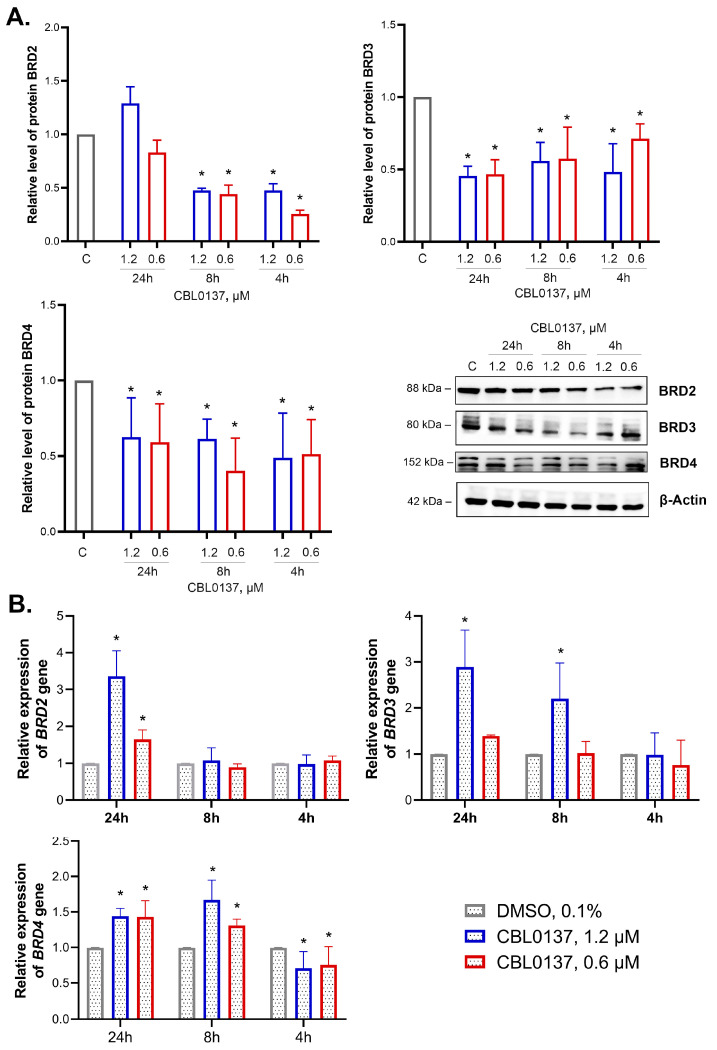
CBL0137 inhibited the expression of BET family proteins and affected BRD2, BRD3 and BRD4 gene expression: (**A**) Expression of BET family proteins BRD2, BRD3, and BRD4 was assessed using Western blotting after treatment of HeLa TI cells with CBL0137 (1.2 and 0.6 μM) for 4 h, 8 h, and 24 h. The results were normalized to β-actin and are shown as mean ± SD of the densitometric analysis (n ≥ 3); * *p* < 0.05 compared to vehicle control (C). (**B**) *BRD2*, *BRD3*, and *BRD4* gene expressions were assessed with RT-qPCR after treatment of HeLa TI cells with CBL0137 (1.2 and 0.6 μM) for 4 h, 8 h, and 24 h. The results were normalized to the housekeeping gene *RPLP0* expression and are shown as mean ± SD (n ≥ 3); * *p* < 0.05 compared to vehicle control (C). The 2^−ΔΔCt^ method was used for data calculation.

**Figure 4 ijms-24-12874-f004:**
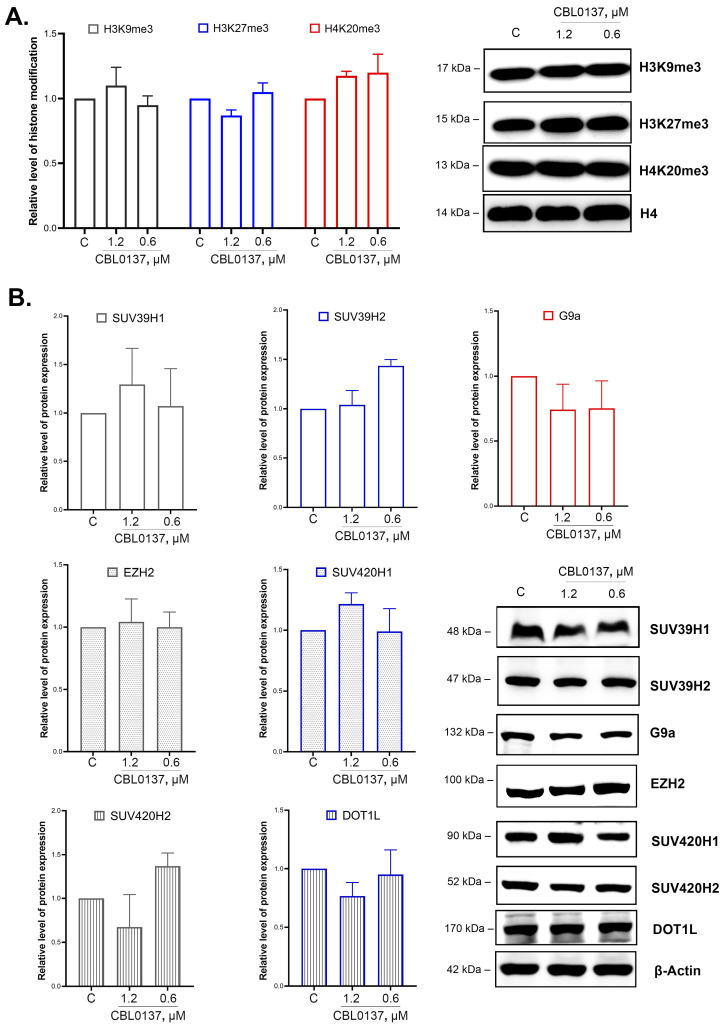
CBL0137 did not affect the integral level of H3K9me3, H3K27me3, and H4K20me3 histone modifications in HeLa TI cells and did not change the expression of HMTs: (**A**) The level of histone modifications H3K9me3, H3K27me3, and H4K20me3 was assessed using Western blotting after treatment of HeLa TI cells with CBL0137 (1.2 and 0.6 μM) for 24 h. The results were normalized to histone H4 and are shown as mean ± SD of the densitometric analysis (n ≥ 3). (**B**) The expression of HMTs SUV39H1, SUV39H2, SUV420H1, SUV420H2, G9a, EZH2, and DOT1L was assessed using Western blotting after treatment of HeLa TI cells with CBL0137 (1.2 and 0.6 μM) for 24 h. The results were normalized to β-actin and are shown as mean ± SD of the densitometric analysis (n ≥ 3).

**Table 1 ijms-24-12874-t001:** Primer sequences.

Gene	Forward Primer (5′-3′)	Reverse Primer (5′-3′)
*DNMT1*	GAGCCACAGATGCTGACAAA	TGCCATTAACACCACCTTCA
*DNMT3A*	AGCCCAAGGTCAAGGAGATT	CAGCAGATGGTGCAGTAGGA
*CREBBP*	CTGGCAGACCTCGGAAAGAA	CTGGCGCCGCAAAAACT
*EP300*	CGCTTTGTCTACACCTGCAA	TGCTGGTTGTTGCTCTCATC
*HAT1*	GCGATAGAGGCACAACAGAA	TGTATTGTTCGGCATCACTCA
*BRD2*	CGGCTTATGTTCTCCAACTGCTA	GGCAGTAGAGACTGGTAAAGGC
*BRD3*	CCAACCATCACTGCAAACGTCAC	GGAGTGGTTGTGTCTGCTTTCC
*BRD4*	CGCTATGTCACCTCCTGTTTGC	ACTCTGAGGACGAGAAGCCCTT
*RPLP0*	CCTTCTCCTTTGGGCTGGTCATCCA	CAGACACTGGCAACATTGCGGACAC

## Data Availability

Data available on request.

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
