# Peer review of "Insights into the Mechanism of Curaxin CBL0137 Epigenetic Activity: The Induction of DNA Demethylation and the Suppression of BET Family Proteins"

_ijms, 2023, doi:10.3390/ijms241612874_

Round 1
Reviewer 1 Report
The study demonstrates that Curaxin CBL0317, a potent DNA-binding small molecule, suppresses DNA methyltransferase DNMT3a and BET family proteins.
The original files of Blots and Gels are not indicated with the condition.
Is it the same bands to the blots in the Figure 3?
The condition and explanation may be added to the blots and gels.
The sentence describing the main goal may be revised to be more clear in lines 76-78.
Author Response
Dear Peer Reviewer,
We are very grateful for your careful reading of our manuscript, its thorough analysis and useful comments. We appreciate your rational pieces of advice very much and did our best to improve our manuscript following your recommendations. Please find our responses on all your comments point by point.
Comment 1: The original files of Blots and Gels are not indicated with the condition. Is it the same bands to the blots in the Figure 3? The condition and explanation may be added to the blots and gels.
Response 1: Following your advice we added separate descriptive files with conditions of the experiments to the original data of all the series of experiments and added captions to the original data files (Supplementary data file). We also indicated files that were used for Figures.
Comment 2: The sentence describing the main goal may be revised to be more clear in lines 76-78?
Response 2: We revised the sentence describing the main goal of our study (lines 82-98). Moreover, following the recommendations of the Reviewer 2 we extended the Introduction (lines 41-44, 55-57, 71-81) and we hope that now the goal of our study become more clear.
Thank you very much for your comments!

Reviewer 2 Report
In this manuscript, the authors investigated the effects of CBL0137 on epigenetic regulators in cancer cell lines. They found that CBL0137 decreased global methylation in Ca Ski cells, and the expression of DNMT3 and BRD2/3/4 proteins in HeLa TI cells. Although the findings are interesting, this study is descriptive, but does not provide any causal relationship between these findings, leaving several questions unanswered as follows:
1. The cytotoxicity of CBL0137 to cell lines should be shown to discuss whether these effects are related to its cytotoxic effect.
2. Since the downregulation of DNMT3A is not examined in Ca Ski cells, it is unclear how CBL0137 decreased global genomic methylation in the cells.
3. Given that genomic methylation is less in HeLa cells innately, it is unlikely that the decrease in genomic methylation is involved in the pharmacological action of CBL0137.
4. It is unclear whether the downregulations of DNMT3A and BRDs are related each other, and are involved in the cytotoxic action of CBL0137.
Round 2
Reviewer 2 Report
The authors have sufficiently addressed my concerns. I have no further comments.